# Differential Effects of Increasing Salinity on Germination and Seedling Growth of Native and Exotic Invasive Cordgrasses

**DOI:** 10.3390/plants8100372

**Published:** 2019-09-25

**Authors:** María Dolores Infante-Izquierdo, Jesús M. Castillo, Brenda J. Grewell, F. Javier J. Nieva, Adolfo F. Muñoz-Rodríguez

**Affiliations:** 1Departamento de Ciencias Integradas. Fuerzas Armadas Ave., Campus El Carmen, Universidad de Huelva, 21071 Huelva, Spain; jimenez@uhu.es (F.J.J.N.);; 2Departamento de Biología Vegetal y Ecología. Universidad de Sevilla, Ap. 1095, 41080 Sevilla, Spain; manucas@us.es; 3USDA-ARS Invasive Species and Pollinator Health Research Unit, Department of Plant Sciences MS-4, 1 Shields Ave., University of California, Davis, CA 95616, USA; bjgrewell@ucdavis.edu

**Keywords:** climate change, dormancy, Odiel Marshes, quiescent seed, salinity tolerance, sea level rise, radicle

## Abstract

Soil salinity is a key environmental factor influencing germination and seedling establishment in salt marshes. Global warming and sea level rise are changing estuarine salinity, and may modify the colonization ability of halophytes. We evaluated the effects of increasing salinity on germination and seedling growth of native *Spartina maritima* and invasive *S. densiflora* from wetlands of the Odiel-Tinto Estuary. Responses were assessed following salinity exposure from fresh water to hypersaline conditions and germination recovery of non-germinated seeds when transferred to fresh water. The germination of both species was inhibited and delayed at high salinities, while pre-exposure to salinity accelerated the speed of germination in recovery assays compared to non-pre-exposed seeds. *S. densiflora* was more tolerant of salinity at germination than *S. maritima*. *S. densiflora* was able to germinate at hypersalinity and its germination percentage decreased at higher salinities compared to *S. maritima*. In contrast, *S. maritima* showed higher salinity tolerance in relation to seedling growth. Contrasting results were observed with differences in the tidal elevation of populations. Our results suggest *S. maritima* is a specialist species with respect to salinity, while *S. densiflora* is a generalist capable of germination of growth under suboptimal conditions. Invasive *S. densiflora* has greater capacity than native *S. maritima* to establish from seed with continued climate change and sea level rise.

## 1. Introduction

Salt marshes are highly stressful environments where halophytes are subjected to high mortality risk [1]. In these habitats, soil salinity is one of the key environmental factors determining vegetation distribution, partially by limiting seed germination and seedling establishment [2]. These phases are crucial in the life cycle of halophytes [3,4,5]. The general behaviour of halophytic seeds in the presence of salt is well documented [6]. Seeds of most halophytes show optimal germination in freshwater, differing in their germination responses to higher salinities [6,7,8]. High salinities usually inhibit germination of halophytes, however, some seeds maintain viability and are able to germinate when osmotic stress decreases [9,10,11].

Estuarine salt marshes are increasingly impacted by biological invasions [12]. Some invasive halophytes show high tolerance to salinity during germination and seedling growth and have colonized a wide range of salt-affected habitats [13,14,15]. In tidal wetlands, climate change and associated sea level rise are changing estuarine salinity patterns [16]. Local environmental conditions can be highly variable with climate change. Salinity decreases in some salt marshes due to an increase in rainfall, while salinity increases in other locations due to sea level rise and increases in temperature and evapotranspiration rates [17,18,19]. These environmental changes may modify the ability of native species to colonize new sites as well as the capacity of invasive halophytes to invade them.

Cordgrasses (genus *Spartina*) provide a model halophyte group to study the responses of native and invasive species to environmental conditions since they inhabit salt marshes around the world, and many species have naturalized in habitats beyond their native ranges [20]. Specifically, native *Spartina maritima* (Curtis) Fernald and invasive *Spartina densiflora* Brongn. co-occur in salt marshes along the Gulf of Cádiz (Southwest Iberian Peninsula) [21]. *S. maritima* is the only native cordgrass in European marshes [22], where it is a primary colonizer at low tidal elevations and facilitates the development of ecological succession [23]. Therefore, the conservation of this species is crucial for the maintenance of biodiversity in these ecosystems.

The effects of salinity on seed germination and seedling growth have never been studied for native *S. maritima*. Actually, seed production of this species has been described as very low or non-existent [22,23,24], but we recently discovered that *S. maritima* in the Southwest Iberian Peninsula produces a moderate number of caryopses (13%) with high variation among tussocks (0%–45%), high viability (89%) and has high germination rates in freshwater (85%) [25]. In contrast, South American *S. densiflora* is one of the three most widely distributed species of the genus and was introduced to the Southwest Iberian Peninsula centuries ago [26]. *S. densiflora* shows high tolerance to environmental variation, including salinity levels [27]. This niche breath has resulted in its colonization of a wide range of different habitats along the intertidal gradient [21,26,28]. Seed production is key to the spread of *Spartina* species [20], and the ability of *S. densiflora* to germinate is a recognized determinant for its invasive potential from brackish marshes to hypersaline saltpans [29,30,31]. To our knowledge, no previous study has evaluated salinity responses of *S. densiflora* seeds that were produced in different habitats along the intertidal gradient. Previous studies of halophytes have found that germination tolerance to salinity depends on environmental conditions in the source habitats where they were produced [9,11,32].

Our main goals were to analyse germination and seedling growth of native *S. maritima* and invasive *S. densiflora* in response to salinity ranging from freshwater to hypersaline levels. We hypothesized that *S. maritima* seeds and seedlings would show high salinity tolerance since the species colonizes low elevation tidal marshes where medium–high salinity levels occur throughout the year [33]. We also hypothesized that *S. densiflora* would show high plasticity in response to salinity since it has invaded a wide range of habitats with contrasted salinity regimes [21]. Germination experiments evaluating the responses of these species to salinity ranging from freshwater to hypersalinity, and recovery of the species after salinity release, were carried out under controlled conditions to test our hypotheses. Results were compared to field conditions where propagules were sourced for the experiments.

## 2. Results

### 2.1. Germination Responses to Salinity

*Spartina maritima* achieved its highest germination percentage (c. 96 %) at low salinity levels between 0.00 and 0.15 M NaCl. Germination rate for the native species decreased at higher salinities. No seed was able to germinate at 0.75 M NaCl. T_50_ G (days necessary to reach 50% of the final germination percentage) was increased at salinities higher than 0.30 M NaCl (Table 1).

Germination percentage for *S. densiflora* decreased significantly in salinities higher than 0.30 M NaCl in seeds from all locations and habitats (Figure 1). However, at higher salinities, the germination percentage of seeds from LM (low marsh) was more than double than those from MM (middle marsh) and HM (high marsh) (Figure 1). Seeds produced in LM showed higher germination (19% ± 3%) than those from MM and HM (c. 9%) at hypersalinity (0.75 M NaCl) (Kruskal–Wallis test, H_2,36_ = 9.16, *p* < 0.05) (Figure 1a–c).

Increasing salinity decreased the germination speed (higher T_50_ G) for seeds sourced from all marsh elevation zones and locations. Additionally, *S. densiflora* seeds produced at LM germinated 27% faster in freshwater than those from MM and HM (one-way ANOVA, F = 11.03, degrees of freedom (df) = 2, *p* < 0.0005) (Table 2).

### 2.2. Germination Responses after Salinity Exposure

The germination rate of *Spartina maritima* seeds increased, and the speed of germination decreased significantly during recovery after exposure to higher salinities. T_50_ R (days necessary to reach 50% of the final germination percentage in recovery assays) was reduced after recovery from all levels of salinity exposure than seeds that were germinated in freshwater. The T_50_ reduction indicating an increase in germination speed was most extreme during recovery from 0.15 M NaCl exposure when germination accelerated almost nine times (Table 1). In contrast to *S. densiflora*, no dormant seeds were recorded for *S. maritima* during the recovery experiment. *S. maritima* seed viability was c. 76%, without showing significant differences among salinities (Table 1). 

As in the case of *S. maritima*, *S. densiflora* seeds had increased germination rates during recovery following exposure to higher salinities, and this result held for seeds sourced from every location and within-marsh habitat elevation zone (Figure 1). *S. densiflora* seeds tended to have increased germination speeds (lower T_50_ R) during recovery following exposure to higher salinities (Table 2). Seeds from LM exposed previously to hypersalinity showed lower recovery germination percentage (78% ± 3%) and lower T_50_ (5 ± 1 days) than seeds from MM and HM (c. 90% and 7 days, respectively) (Figure 1, Table 2). Dormant *S. densiflora* seeds in freshwater were c. 7% for every habitat and location and tended to decrease with increasing salinity exposure (Figure 1). Seed viability in control treatments were higher for seeds of *S. densiflora* from LM and MM (c. 76%) than those from HM (54% ± 4%) (one-way ANOVA, F = 12.95, df = 2, *p* < 0.0001). On the other hand, seed viability was not affected by salt treatments in seeds from LM, while in seeds from MM and HM the viability decreased as salinity increased (Table 2). Regarding the three source population locations, there were no significant differences in seed viability at any salinity level for seed sourced from Almendral, whereas viability was reduced at higher salinities for the other two study locations (Table 2).

### 2.3. Initial Seedling Growth Responses to Salinity

*S. maritima* seedlings had similar cotyledon and first leaf lengths at every salinity level (Figure 2). However, the radicles of *S. maritima* were four times longer at 0.15 M NaCl than radicles that emerged from seeds exposed to other salinity concentrations (Kruskal–Wallis test, H_3,59_ = 8.66, *p* < 0.05). Cotyledon, first leaf and radicle of *S. maritima* seedlings growing in freshwater all increased in length in recovery after seeds had been exposed to salinity higher than 0.15 M NaCl (Figure 2).

The cotyledon and first leaf length of *S. densiflora* were reduced as salinity increased in seeds sourced from every study location and elevational habitat (Figure 3). The length of cotyledon and first leaf were shorter for seedlings from LM seed that germinated at salinity higher than 0.30 M NaCl (Figure 3a). In contrast, seedlings from MM and HM seeds displayed this reduction in size at a lower salinity concentration of 0.15 M NaCl (Figure 3b,c). Also, in contrast, salinity had a positive effect on radicle growth at 0.30 M NaCl for seeds coming from LM, though radicle length was reduced at salinities higher than 0.45 M NaCl (Kruskal–Wallis test, H_5,205_ = 84.93, *p* < 0.0001) (Figure 3a). This shorter radicle length trait was expressed in salinities higher than 0.30 M NaCl for seeds from MM (Kruskal–Wallis test, H_5,147_ = 28.99, *p* < 0.0001) (Figure 3b) and higher than 0.15 M NaCl for seeds from HM (Kruskal–Wallis test, H_5,140_ = 26.65, *p* < 0.001) (Figure 3c). In freshwater conditions, *S. densiflora* seedlings emerging from seeds sourced at Almendral and Bacuta locations had 1.3 times, and 1.7 times larger first leaves and radicles respectively, than those from Calatilla (Kruskal–Wallis test, H_2,111_ = 10.17, *p* < 0.01; one-way ANOVA test, F = 4.88, df = 2, *p* < 0.01, respectively) (Figure 3d–f). Few significant differences, without showing a clear pattern, were recorded for seedling responses during the recovery experiment for all previous salinity exposures (Figure 3).

## 3. Discussion

Our hypotheses were partially confirmed by the outcomes of the experiments. As expected, germination rates of seeds from native *S. maritima* expressed high tolerance to salinity, but tolerance of this native species was lower than that of salinity levels tolerated by invasive *S. densiflora*. In this circumstance, *S. densiflora* has a higher capacity for phenotypic plasticity in response to salinity. However, invasive seedling trait responses to salinity suggest a lower tolerance to salinity during the initial 15 days of seedling growth than the more vigorous initial growth of native *S. maritima*.

Some salinity responses were common to both studied cordgrasses. For example, high seed viability after salinity exposure was recorded for both *Spartina* species, as it has commonly been observed for many halophyte species [9]. Also, elevated salinity concentrations inhibited and delayed germination for both studied *Spartina* species [8]. This seed quiescence prevents seed germination under stressful conditions [1,5]. Decreases in germination rates have been reported for the congener *Spartina alterniflora* Loisel. at salinities higher than 0.20–0.40 M NaCl in the native and invasive range [34,35], however some authors recorded high germination (>90%) even at hypersalinity [15]. Germination of *Spartina ciliata* Brongn. from Brazil was reduced at salinities higher than 0.20 M NaCl, totally inhibited at seawater concentration, and speed of germination increased after salinity exposure [36]. In addition, exposure to salinity followed by recovery after freshening accelerated germination for *S. maritima* and *S. densiflora* in our study, which has been reported previously for other halophyte species from a range of functional groups [8,37,38,39]. The observed stimulation of germination speed and rate after salinity exposure can provide windows of opportunity for seeds to germinate and quickly establish when salinity is sporadically and temporarily reduced by precipitation events or other sources of freshwater inflow [10,40].

While both studied cordgrasses showed high tolerance to salinity and shared some common germination and initial seedling trait responses, each also expressed distinctly contrasting responses. Germination rates and speed of germination under increasing NaCl concentrations indicated *S. densiflora* had higher germination tolerance to salinity than *S. maritima*. *S. maritima* germination was completely inhibited and *S. densiflora* was able to germinate at hypersalinity. Moreover, germination percentage decreased from 0.15 M NaCl up for *S. maritima* and from 0.30 M NaCl up for *S. densiflora*. As in our study, invasive *S. densiflora* in Humboldt Bay (California) showed reductions in seed germination at salinities higher than 0.30 M NaCl [30]. Some authors recorded total germination inhibition at 1.00 M NaCl and at 0.70 M NaCl for invasive *S. densiflora* in the Gulf of Cádiz [29,31]. The great salinity tolerance of *S. densiflora* was also reflected on its rapid germination after being pre-treated at increasing salinities, whereas this study is the first to document the opposite response for native *S. maritima*. Seed quiescence of *S. densiflora* did not alter its initial seedling growth, in contrast to responses of *S. maritima,* in which seedlings were longer after saline pre-treatments compared to control. In addition to the capacity for seed quiescence under stressful conditions, the response of *S. densiflora* indicates a degree of physiological seed dormancy ( < 10%). In addition, the radicle and first leaf of *S. densiflora* always emerged earlier than those of *S. maritima*. This suggests *S. densiflora* seeds were able to remain dormant in stressful saline environments without damaging the quality of the embryo, and then had the capacity to germinate later when salinity stress was reduced, providing multiple opportunities for establishment [1,10,39].

Its higher germination tolerance to salinity, faster germination after being pre-treated at increasing salinities and the presence of seed dormancy help to explain that *S. densiflora* is able to invade a wide range of habitats along the intertidal gradient [21] including hypersaline saltpans [3].

Invasive *S. densiflora* germination showed higher salinity tolerance than native *S. maritima*, but the autochthonous species showed higher salinity tolerance in relation to early seedling growth. Negative effects of salinity on seedling growth have been reported previously for invasive *S. densiflora* in Humboldt Bay (California) where seedling height decreased at salinities higher than 0.20 M NaCl [30], for invasive *S. alterniflora* in China [35], where shoot height decreased from 0.20 M NaCl up and radicle length from 0.10 M NaCl up, and for native *S. ciliata* in Brazil [36], with shoot and radicle being smaller at salinities higher than from 0.05 M NaCl. 

Besides the general comparison between both *Spartina* species, contrasted responses to salinity in seed viability, germination rate and speed, seed dormancy and seedling growth were also recorded among native and invasive species and among *S. densiflora* populations along the intertidal gradient. These differences in germination and seedling trait responses between *S. maritima* and *S. densiflora* and for seeds sourced among contrasting *S. densiflora* habitats could be attributed to local adaptation to contrasted environments [41], or to pre-adaptive conditioning determined by the maternal environment during seed development [42,43,44]. Salinity responses from different *S. densiflora* locations along the Odiel-Tinto Estuary (grouping LM, MM and HM elevations at each location) may support the pre-adaptive conditioning hypothesis. Supporting this idea, seed viability was high and independent of exposure to salinity concentrations for seeds from Almendral, but it decreased as salinity increased for seeds from Bacuta and Calatilla locations. Almendral is the nearest location to the coastline (12,500 m), whereas Bacuta and Calatilla are located along a tidal gradient 1800 m and 5500 m inland from Almendral, respectively. Thus, seeds ripening in low elevations and in locations closer to the coastline are more frequently exposed to tidal flooding and salt spray than those at higher elevations and more inland locations on the intertidal gradient [45]. Differentiated environmental conditions may acclimate seeds to salt stress in LM and closer to the sea, protecting their embryo from being killed due to ion toxicity at high salinities [2]. Furthermore, the invasive *S. densiflora* populations have low genetic diversity in North American and European marshes [46,47], which also supports that differences recorded along the intertidal gradient would likely be due to phenotypic plasticity rather than to genetic adaptation. Pre-adaptive conditioning determined by maternal stress conditions could increase survivorship and germination under high salinities, which may suppose an advantage for offspring in conditions similar to those experienced by the parents [42,43,44]. Moreover, salinity acclimation would facilitate survivorship during hydrochorus dispersal of buoyant seeds with sea water currents [48]. Other authors have observed that salt tolerance in halophyte germination is related to the duration and intensity of their exposure to salts in field conditions [9,11,32,49]. As in our study, *Iris hexagona* Walter and *Suaeda aralocaspica* (Bunge) Freitag and Schütze growing in high salinities produced seeds that had higher germination rates and speeds of germination when seeds were exposed to different salinity concentrations than seeds produced in low salinity environments [37,50]. Furthermore, other environmental factors such as temperature, photoperiod, soil moisture and nutrients availability can influence seed viability, germinability and dormancy [1,51,52,53,54].

## 4. Materials and Methods

### 4.1. Study Area and Plant Material

The plant propagules evaluated in this study were sourced from the Odiel Marshes by the Gulf of Cádiz, in the Southwest Iberian Peninsula. The coast of the Gulf of Cádiz is mesotidal and the mean sea level in this area is +1.85 m relative to Spanish Hydrographic Zero (SHZ). The tides are semidiurnal and have a mean range of 2.10 m and a mean spring tidal range of 2.97 m, representing 0.40–3.37 m above SHZ. This area is under a Mediterranean climate with Atlantic influence, with +18.2 °C as the annual mean temperature [23]. Native vegetation in salt marshes along the Gulf of Cádiz has been described in previous works [23,33,55].

Inflorescences in fruiting stage were randomly collected from *S. maritima* and *S. densiflora* tussocks in the Odiel Marshes. Since native *S. maritima* colonizes mainly low elevations in the tidal frame [23], inflorescences were collected in August 2017 from a low marsh at the location known locally as Ludovico (37.174341N, –6.931643W; See a site description in previous work [23]) (Figure 4). *S. densiflora* invades LM, MM and HM in the Gulf of Cádiz [21], so its inflorescences in fruiting stage were collected from those three habitats at each of three different locations distributed from close to the estuary inlet to more inland areas (Almendral: 37.209699N, –6.953506W; Bacuta: 37.218836N, −6.964066W; Calatilla: 37.250382N, −6.969434W) in November 2016 (Figure 4). Marsh habitats were distinguished based on tidal influence and soil characteristics [33]. LM were defined between Mean High-Water Neap (MHWN) and Mean High Water (MHW), MM went from Mean High Water (MHW) to Mean High Water Spring (MHWS), and HM from Mean High Water Spring (MHWS) to Highest Astronomical Tide (HAT) [56]. In all sampled salt marshes, *S. densiflora* has become very abundant, displacing native vegetation [21]. Spikelets containing caryopses were randomly selected from collected inflorescences and stored in paper bags in dark and dry conditions at +5 °C until use.

### 4.2. Salinity Germination Experiment

Before sowing, in September 2017 in the case of *S. maritima* and in December 2016 in the case of *S. densiflora*, the spikelets of both *Spartina* species were surface sterilized in 5% (v/v) sodium hypochlorite for 10 min to prevent fungal contamination and then rinsed with distilled water [25,39]. Four replicates with 25 spikelets each were sown, for each habitat and location, on two layers of autoclaved filter paper watered with six different salt treatments (sodium chloride puriss pro analysis >99.5%, Sigma-Aldrich; 0.00 (control), 0.15, 0.30, 0.45, 0.60 and 0.75 M NaCl) in Petri dishes (9 cm diameter) sealed with adhesive tape (Parafilm™) to avoid desiccation. Sodium chloride was chosen as the salt to be investigated since it is by far the most prevalent major salt dissolved in the Odiel estuary water [57]. This salinity range was chosen to include salinities from freshwater (0.0 M NaCl) to sea water (0.60 M NaCl), and hypersalinity (0.75 M NaCl). The dishes were maintained during 2 months under controlled-environmental conditions in a plant grow room, at temperatures between 20 °C and 25 °C and a 12 h light/12 h dark photoperiod. Radiation was provided by fluorescent lamps that produced a photosynthetic photon flux density of 60 μmol m^−2^ s^−1^. During this time, germination was recorded every 2 or 3 days. A seed was considered germinated when the coleoptile emerged.

### 4.3. Post-Salinity Exposure Recovery Experiment

Spikelets that did not germinate during the 2 months salinity exposure trials were rinsed with distilled water and sown in new Petri dishes with distilled water to assess post-salinity exposure recovery. Germination was recorded every 2 or 3 days for 2 months. Seed viability of the spikelets that did not germinate during the recovery experiment was tested using the Tetrazolium test [58]. For this purpose, the embryo was incised with a scalpel and submerged in a 1% aqueous solution of 2,3,5 triphenyl tetrazolium chloride at 25 °C in darkness for 24 h. Then, red-stained viable embryos were counted through a magnifying glass. 

The percentage of viable seeds (germinated plus dormant seeds) was calculated for each Petri dish. The germination rates (percentage) for viable seeds at different salinities, and the recovery germination percentage after salt exposure were then calculated. Seeds that did not germinate during the salinity treatments, but germinated in the recovery experiment, were considered quiescent seeds. Seed dormancy percentage was calculated for each Petri dish using the number of viable seeds that did not germinate at the end of the recovery experiment. In addition, the days necessary to reach 50% of the final germination percentage was calculated for each Petri dish in both the germination experiment (T_50_ G) and the recovery experiment (T_50_ R) [25,39].

### 4.4. Initial Seedling Growth

To evaluate the effects of salinity exposure and post-salinity recovery on initial seedling growth, the cotyledon, first leaf and radicle length of 1–7 seedlings per Petri dish (n = 4 Petri dishes per treatment) were measured under a magnifying glass using a ruler [59]. These data were recorded 15 days after germination in both experiments to assess initial growth of both *S. maritima* and *S. densiflora* seedlings.

### 4.5. Statistical Analysis

Statistical analyses were carried out with STATISTICA 8.0 (StatSoft Inc., USA) considering significant results when *p* ≤ 0.05. Deviation to the mean was calculated as Standard Error (SE). The normality of the data series was tested with Kolmogorov–Smirnov test and the homogeneity of variance using the Levene test. When data or their transformations (using $\sqrt{x}$, 1∕(x+1) or arcsine(x) functions) had a normal distribution and presented homeostasie, differences in germination parameters and seedling measurements between different salinities were analysed using one-way analysis of variance (ANOVA) and Tukey’s Honest Significant Difference (HSD) test as post-hoc test. If data series did not have a normal distribution or homogeneity of variance after transformation, we evaluated response differences using a non-parametric Kruskal–Wallis H-test and a Mann–Whitney U post-hoc test.

## 5. Conclusions

Together, our results provide new information on seed germination and early seedling life stage characteristics of native *Spartina maritima* in comparison to responses of co-occurring invasive South American *S. densiflora* to increasing estuarine salinity changes driven by global warming and sea level rise. At these life stages, critical to survival and establishment, *S. maritima* displayed a specialist strategy by germinating primarily under salinity concentrations that support survival and optimal initial seedling growth. This strategy is in accordance with field observations with *S. maritima* colonizing mostly only stressful low salt marshes. In contrast, invasive *S. densiflora* behaved as a generalist species [21] and showed the capacity to germinate and produce seedlings under a wide range of salinity concentrations. However, it presented its optimum seedling growth at freshwater and light brackish conditions, with sub-optimum seedling growth at higher salinities. This behaviour is in agreement with *S. densiflora* adult individuals showing high phenotypic plasticity for many traits and opportunistically colonizing a wide range of habitats along the intertidal gradient, though often as sub-optimal phenotypes for the conditions [46]. In view of these results, invasive *S. densiflora* seems to be better prepared than native *S. maritima* to tolerate salinity changes provoked by climate change and sea level rise [16]. Conservation priorities to provide a future habitat for *S. maritima* and other native tidal wetland flora should consider preservation of undeveloped uplands for accommodation space above current high water levels for estuarine marsh transgression with sea level rise, and immediate implementation of invasive plant management, as successional development of new tidelands will favour aggressive alien colonizers such as *S. densiflora* [60]. Considering this work, wetland restoration strategies should consider seed and seedling life stage responses under changing environmental conditions to support recruitment and establishment of *S. maritima*.

## Figures and Tables

**Figure 1 plants-08-00372-f001:**
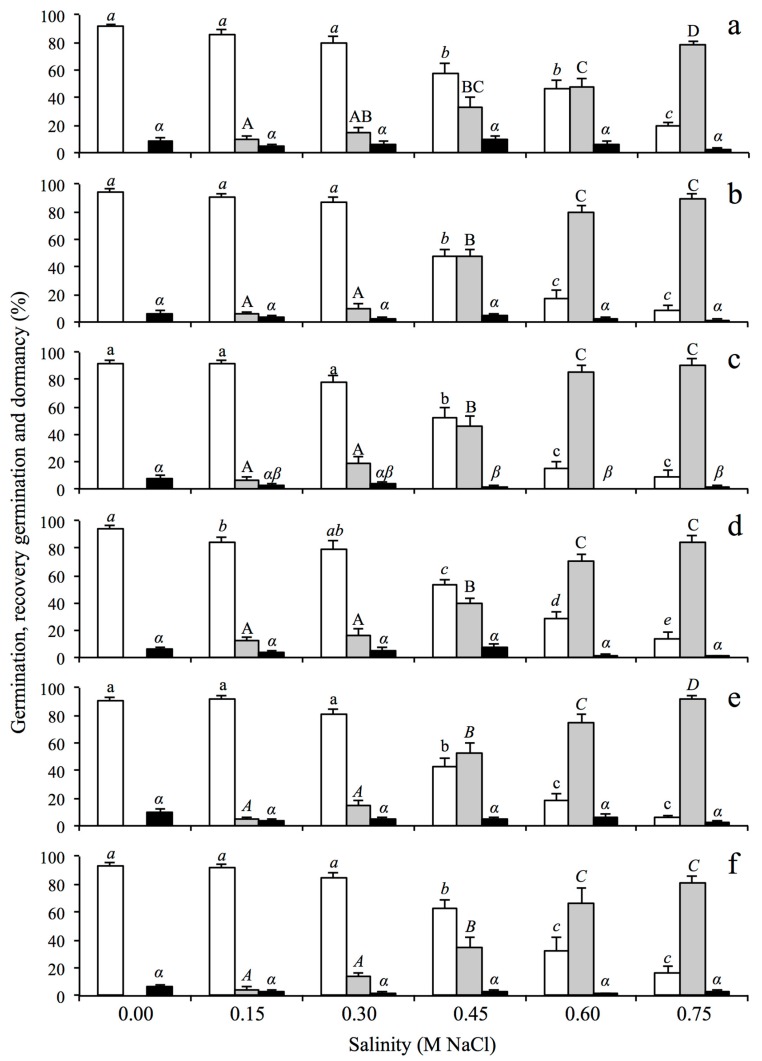
Percentages of germination (white bars), recovery of germination after salt exposure (grey bars) and seed dormancy (black bars) in six salinity treatments for *Spartina densiflora* from three habitats (grouping the three locations in each habitat): (**a**) low marsh (LM), (**b**) middle marsh (MM), (**c**) high marsh (HM), and from three locations (grouping the three habitats in each location): (**d**) Almendral, (**e**) Bacuta, (**f**) Calatilla, in the Odiel Marshes (Southwest Iberian Peninsula). Data are mean ± SE (n = 4). Different letters indicate significant differences among treatments for each trait (Mann–Whitney U test, in italic, for Kruskal–Wallis or Tukey’s HSD test, in non-italic, for one-way ANOVA, *p* < 0.05).

**Figure 2 plants-08-00372-f002:**
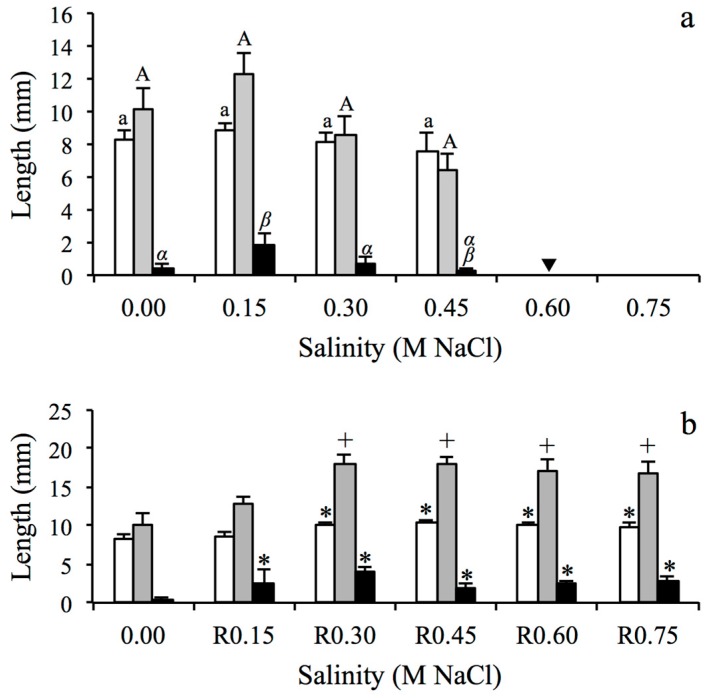
Cotyledon (white bars), first leaf (grey bars) and radicle length (black bars) for seedlings of *Spartina maritima* germinated: (**a**) in six salt treatments and (**b**) in the recovery (R) assays after salinity exposure. Data show mean ± SE (n = 5–13). Different letters indicate significant differences between salinity treatments (Mann–Whitney U test, in italic, for Kruskal–Wallis or Tukey’s HSD test, in non-italic, for one-way ANOVA, *p* < 0.05). Asterisks and plus sign indicate significant differences compared to control treatment (0.00 M) for each seedling parameter (Mann–Whitney U test, asterisks, for Kruskal–Wallis or Tukey’s HSD test, plus sign, for one-way ANOVA, *p* < 0.05). Triangle indicates that seedlings were dead before 15 days after germination, so they were not measured.

**Figure 3 plants-08-00372-f003:**
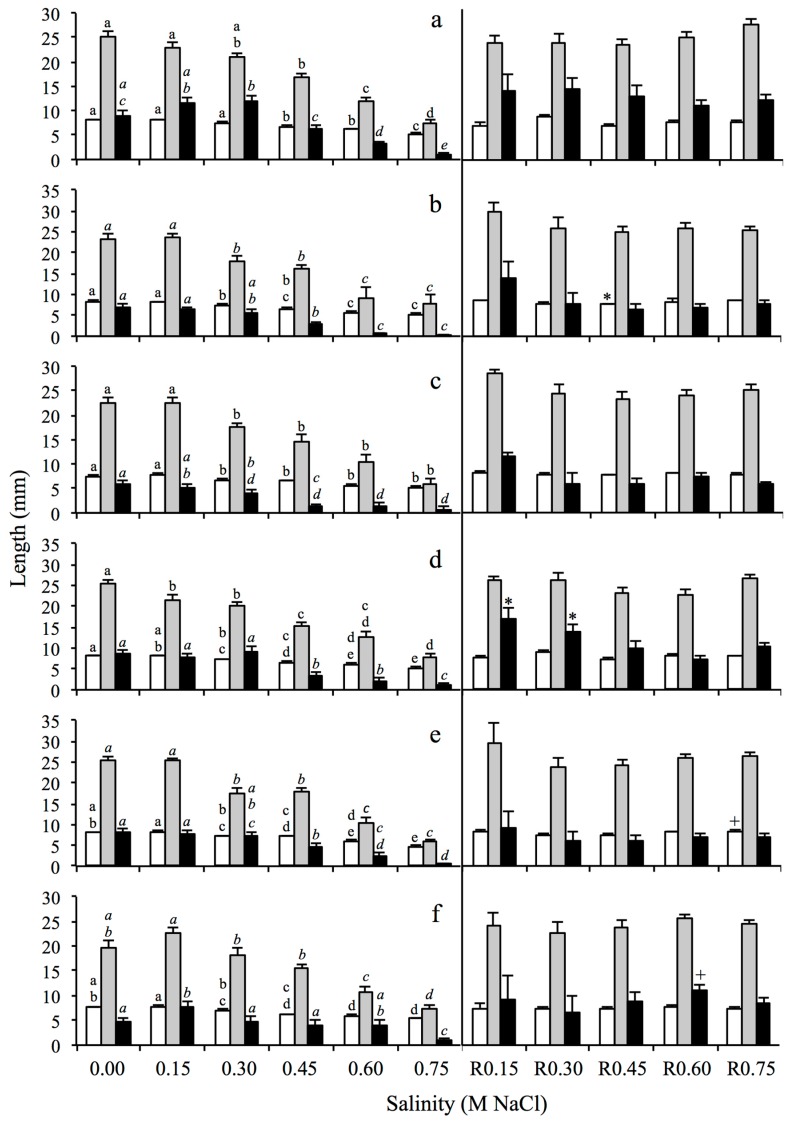
Cotyledon (white bars), first leaf (grey bars) and radicle length (black bars) from seedlings of *Spartina densiflora* germinated in six salinity treatments (left) and in the recovery assays after salinity exposure (R) (right) coming from seeds collected from three habitats (grouping the three locations in each habitat): (**a**) low marsh, (**b**) middle marsh, (**c**) high marsh, and from three locations (grouping the three habitats in each location): (**d**) Almendral, (**e**) Bacuta. (**f**) Calatilla), in the Odiel Marshes (Southwest Iberian Peninsula). Data show mean ± SE (n = 3–40). Different letters indicate significant differences among salinity treatments (Mann–Whitney U test, in italic, for Kruskal–Wallis or Tukey’s HSD test, in non-italic, for one-way ANOVA, *p* < 0.05). Asterisks and plus sign indicate significant differences compared to control treatment (0.00 M) for each seedling parameter (Mann–Whitney U test, asterisks, for Kruskal–Wallis or Tukey’s HSD test, plus sign, for one-way ANOVA, *p* < 0.05).

**Figure 4 plants-08-00372-f004:**
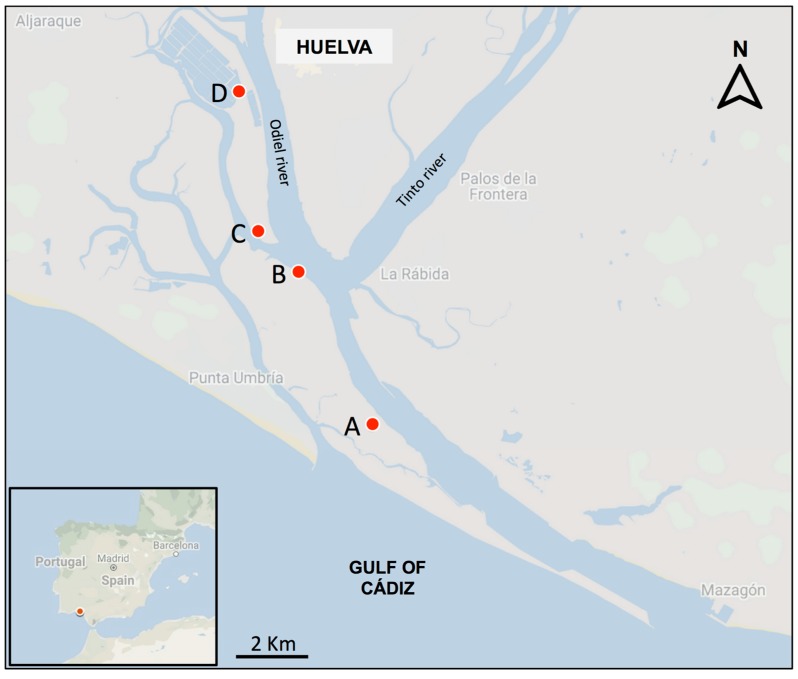
Map of the Odiel Marshes (Southwest Iberian Peninsula) showing (**A**) the location where inflorescences in the fruiting stage of native *Spartina maritima* were collected from a low marsh, and the three locations (**B**, Almendral; **C**, Bacuta; **D**, Calatilla) where inflorescences of invasive *S. densiflora* were collected from low, middle, and high marshes. (Source: Google Maps, data from ©2019 Instituto Geográfico Nacional Spain).

**Table 1 plants-08-00372-t001:** Germination percentage (G), T_50_ of germination (T_50_ G), recovery germination percentage after salt exposure (RG), T_50_ of germination recovery (T_50_ R), and seed viability percentage (V) for native *Spartina maritima* from the Gulf of Cádiz (Southwest Iberian Peninsula) in six salt treatments. Data show mean ± Standard Error (SE) (n = 3–4). Different letters indicate significant differences between treatments (Mann–Whitney U test, in italic, for Kruskal–Wallis or Tukey’s Honest Significant Difference (HSD) test, in non-italic, for Analysis of Variance (one-way ANOVA), *p* < 0.05). df (degrees of freedom).

Salinity (M NaCl)	G (%)	T_50_ G (days)	RG (%)	T_50_ R (days)	V (%)
0.00	100 ± 0a	23 ± 1a	-	-	76 ± 6a
0.15	91 ± 4a	28 ± 2ab	9 ± 4a	3 ± 1*a*	76 ± 8a
0.30	51 ± 6b	32 ± 4abc	49 ± 6b	6 ± 1*a*	76 ± 7a
0.45	19 ± 4c	38 ± 6bc	81 ± 4c	11 ± 1*b*	78 ± 1a
0.60	12 ± 5cd	43 ± 3c	88 ± 5cd	12 ± 1*b*	71 ± 3a
0.75	0 ± 0d	-	100 ± 0d	13 ± 0*b*	78 ± 3a
one-way ANOVA (F) or Kruskal–Wallis (H) test	F = 119.17, df = 5, *p* < 0.0001	F = 5.43, df = 4, *p* < 0.01	F = 74.59, df = 4, *p* < 0.0001	H_4,19_ = 14.64, *p* < 0.01	F = 0.22, df = 5, *p* > 0.05

**Table 2 plants-08-00372-t002:** Comparisons among salt treatments for *Spartina densiflora* in the Southwest Iberian Peninsula. Percentage of viability (V), T_50_ of germination (T_50_ G) and T_50_ of recovery (T_50_ R) in the different salt treatments (0.00, 0.15, 0.30, 0.45, 0.60 and 0.75 M NaCl) for different habitats and locations. Data are mean ± SE (n = 12). Different letters indicate significant differences among treatments for each trait (Mann–Whitney U test, in italic, for Kruskal–Wallis or Tukey’s HSD test, in non-italic, for one-way ANOVA, *p* < 0.05). df (degrees of freedom).

Salinity (M NaCl)	V (%)	T_50_ G (days)	T_50_ R (Days)	V (%)	T_50_ G (days)	T_50_ R (days)
		**Low marsh**		**‘Almendral’**		
0.00	79 ± 3a	19 ± 1a	-	65 ± 6a	22 ± 1a	-
0.15	82 ± 3a	22 ± 1ab	24 ± 5*a*	65 ± 6a	25 ± 1ab	20 ± 4*a*
0.30	80 ± 3a	25 ± 2b	15 ± 3*ab*	62 ± 7a	27 ± 2abd	12 ± 3*ab*
0.45	79 ± 4a	27 ± 2bc	9 ± 1*bc*	57 ± 8a	32 ± 3bc	8 ± 1*bc*
0.60	73 ± 4a	33 ± 3c	6 ± 1*cd*	55 ± 7a	40 ± 4c	5 ± 1*c*
0.75	71 ± 3a	36 ± 2c	5 ± 1*d*	52 ± 8a	39 ± 4cd	6 ± 0*bc*
one-way ANOVA (F) or Kruskal–Wallis (H) test	F = 1.77, df = 5 *p* > 0.05	F = 14.16, df = 5 *p* < 0.0001	H_4,55_ = 17.51 *p* < 0.005	F = 0.61, df = 5 *p* > 0.05	F = 9.38, df = 5 *p* < 0.0001	H_4,53_ = 19.85 *p* < 0.001
	**Middle marsh**			**‘Bacuta’**		
0.00	70 ± 4a	26 ± 1*a*	-	66 ± 5a	23 ± 1*a*	-
0.15	64 ± 2ab	28 ± 2*ab*	11 ± 2*a*	63 ± 5a	27 ± 2*ab*	19 ± 6*a*
0.30	56 ± 2bc	31 ± 2*ab*	9 ± 3*a*	57 ± 6a	28 ± 2*b*	13 ± 3*a*
0.45	45 ± 3cd	37 ± 4*bc*	12 ± 4*a*	44 ± 6a	31 ± 3*b*	13 ± 4*a*
0.60	41 ± 3d	39 ± 8*abc*	7 ± 1*a*	47 ± 6a	31 ± 5*abc*	7 ± 1*a*
0.75	39 ± 3d	43 ± 5*c*	7 ± 0*a*	46 ± 5a	43 ± 4*c*	6 ± 1*a*
one-way ANOVA (F) or Kruskal–Wallis (H) test	F = 18.08, df = 5 *p* < 0.0001	H_5,62_ = 13.71 *p* < 0.05	H_4,51_ = 5.08 *p* > 0.05	F = 2.91, df = 5 *p* < 0.05	H_5,63_ = 20.42 *p* < 0.005	H_4,51_ = 6.27 *p* = 0.180
		**High marsh**		**‘Calatilla’**		
0.00	54 ± 4a	25 ± 1*a*	-	71 ± 3a	24 ± 2*a*	-
0.15	46 ± 3ab	25 ± 1*a*	31 ± 9*a*	64 ± 3a	24 ± 2*a*	25 ± 8*a*
0.30	37 ± 3bc	28 ± 2*ab*	13 ± 3*ab*	54 ± 5ab	29 ± 3*ab*	11 ± 3*a*
0.45	32 ± 3c	36 ± 4*abc*	7 ± 1*b*	55 ± 5ab	38 ± 5*ab*	7 ± 1*a*
0.60	30 ± 2c	37 ± 5*bc*	6 ± 1*b*	43 ± 5b	37 ± 5*b*	6 ± 1*a*
0.75	30 ± 3c	42 ± 6*c*	7 ± 1*b*	42 ± 5b	33 ± 1*b*	6 ± 1*a*
one-way ANOVA (F) or Kruskal–Wallis (H) test	F = 10.14, df = 5 *p* < 0.0001	H_5,57_ = 16.94 *p* < 0.005	H_4,49_ = 14.91 *p* < 0.005	F = 6.81, df = 5 *p* < 0.0001	H_5,62_ = 12.99 *p* < 0.05	H_4,51_ = 6.21 *p* > 0.05

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
