# Peer review of "Differential Effects of Increasing Salinity on Germination and Seedling Growth of Native and Exotic Invasive Cordgrasses"

_plants, 2019, doi:10.3390/plants8100372_

Round 1

Reviewer 1 Report

This interesting work describes the effects of salinity on the germination and growth of one native and one invasive variety of cordgrass, a halophyte grass. The paper is laid out well and the methods, results, data analysis (including statistical analysis) and conclusions are clearly presented. There are a few suggestions I have for improving the manuscript, as follows:

Line 21 in abstract, the sentence is a little confusing. "...pre-exposure to salinity accelerates germination..." Compared to what? I assume compared to sudden salinity shock but it could also be taken as compared to no salinity; but this would contradict the first part of the sentence. Please clarify.

Line 40, insert the word 'to' so it reads "and are able to germinate..."

Line 73, insert the word 'to' so it reads "...in response to salinity..."

Lines 94-98 and 135-139, Both of these sentences are quite long and confusing, and could be split into two sentences or reworded to make more clear.

Line 226, delete 's' so it reads "...seedling growth..."

Author Response

We thank you for your comments.

Point 1: Line 21 in abstract, the sentence is a little confusing. "...pre-exposure to salinity accelerates germination..." Compared to what? I assume compared to sudden salinity shock but it could also be taken as compared to no salinity; but this would contradict the first part of the sentence. Please clarify.

Response 1: We have changed the sentence to:

“Responses were assessed following salinity exposure from fresh water to hypersaline conditions and germination recovery of non-germinated seeds when transferred to fresh water. The germination of both species was inhibited and delayed at high salinities, while pre-exposure to salinity accelerated the speed of germination in recovery assays compared to non pre-exposed seeds.”

Point 2: Line 40, insert the word 'to' so it reads "and are able to germinate..."

Response 2: “and are able germinate...” has changed to “and are able to germinate...”

Point 3: Line 73, insert the word 'to' so it reads "...in response to salinity..."

Response 3: “...in response salinity...” has changed to “...in response to salinity...”

Point 4: Lines 94-98 and 135-139, Both of these sentences are quite long and confusing, and could be split into two sentences or reworded to make more clear.

Response 4:

L94-98: Germination percentage for S. densiflora decreased significantly in salinities higher than 0.30 M NaCl for seed sourced from all study locations and habitats within locations (ANOVA or Kruskal-Wallis test, p < 0.0001), although this reduction was less pronounced for LM (low marsh) seeds that showed more than double germination percentage than MM (middle marsh) and HM (high marsh) seeds at salinities higher than 0.45 M NaCl (Figure 1).

These sentences have changed to:

"Germination percentage for S. densiflora decreased significantly in salinities higher than 0.30 M NaCl in seeds from all locations and habitats (ANOVA or Kruskal-Wallis test, p < 0.0001). However, at higher salinities the germination percentage of seeds from LM (low marsh) was more than double than those from MM (middle marsh) and HM (high marsh) (Figure 1)".

L135-139: S. densiflora seed viability at freshwater was higher for seeds sourced from LM and MM (c. 76%) than in HM (54 ± 4%) (ANOVA, F = 12.95, df = 2, p < 0.0001). Seed viability was similar in response to every tested salinity level for seeds produced at LM, but viability decreased in freshwater to 0.45 M NaCl ranges for seeds coming from MM and HM (Table 2).

These sentences have changed to:

"Seed viability in control treatments were higher for seeds of S. densiflora from LM and MM (c. 76%) than those from HM (54 ± 4%) (ANOVA, F = 12.95, df = 2, p < 0.0001). On the other hand, seed viability was not affected by salt treatments in seeds from LM, while in seeds from MM and HM the viability decreased as salinity increased (Table 2)".     

Point 5: Line 226, delete 's' so it reads "...seedling growth..."

Response 5: “...seedlings growth...” has changed to “...seedling growth...”

Reviewer 2 Report

Natural resources in the world are becoming increasingly limited, as we speak of biotic or abiotic resources. In this context, the decrease of fresh water can determine the occurrence of phenomena of salting of the soil and implicitly of adapting the plants to new environmental conditions. Thus, soil salinity influences not only seed germination but also plant growth and development with negative implication on biodiversity.

In the Introduction – is necessary a phrase about the importance of maintaining the biodiversity of the Spartina genus in the context of the research area

In the tables and figures it is necessary to specify concretely which statistical test was used Manu-Whitney U test or Tukey HSD test.

Line 100 - Figure 1 must be moved into the text where reference is made  In the Table 1 must be presented df (degree of freedom)

Anova must be write ANOVA with Caps lock (Table 1, 2, 3,)

In the text must be indicated what statistical calculation was used ANOVA or Kruskal-Wallis test 

Line 186 – change As cojuctured with In this circumstance

Lines 271 and 275 change  state word with stage or phenophase

Line 338 Tukey test is same a non parametric test

Conclusion:

Line 352-354 did not use references

The font in the figures 1,2,3 must be TNR

Author Response

We thank you for your comments.

Point 1: In the Introduction – is necessary a phrase about the importance of maintaining the biodiversity of the Spartina genus in the context of the research area

Response 1: A phrase was added in the Introduction:

S. maritima is the only native cordgrass in European marshes [22], where it is a primary colonizer at low tidal elevations and facilitates the development of ecological succession [23]. Therefore, the conservation of this species is crucial for the maintenance of biodiversity in these ecosystems”.

Point 2: In the tables and figures it is necessary to specify concretely which statistical test was used Manu-Whitney U test or Tukey HSD test.

Response 2: We have used italic and non-italic letters for the different test in tables and figures, and we have described it in figure captions and in table headers.

Point 3: Line 100 - Figure 1 must be moved into the text where reference is made  

Response 3: Figure 1 has been moved.

Point 4: In the Table 1 must be presented df (degree of freedom)

Response 4: “df, degrees of freedom” has changed to “df (degrees of freedom)” in Table 1, and was added in Table 2.

Point 5: Anova must be write ANOVA with Caps lock (Table 1, 2, 3,)

Response 5: Anova has changed to ANOVA in Tables.

Point 6: In the text must be indicated what statistical calculation was used ANOVA or Kruskal-Wallis test 

Response 6: As commented in point 2, we have indicated in tables and figures the specific test used in each comparison. Indications in text have been changed by references to tables or figures.

Point 7: Line 186 – change As cojuctured with In this circumstance

Response 7: “As conjectured” has changed to “In this circumstance”

Point 8: Lines 271 and 275 change state word with stage or phenophase

Response 8: The word “state” has changed to “stage” in text and in the Figure 4 caption

Point 9: Line 338 Tukey test is same a non parametric test

Response 9: Yes, but experts told us that Kruskal-Wallis test analyzes rank-distributions, while Tukey's HSD is about means, and they told us that it is not appropriate to mix different hypotheses.

Point 10: Line 352-354 did not use references

Response 10: Reference has been added: "In contrast, invasive S. densiflora behaved as a generalist species [21],..."

Point 11: The font in the figures 1,2,3 must be TNR

Response 11: The font in Figures 1, 2 ,3 has been changed to TNR

Reviewer 3 Report

The authors are presenting an interesting paper. However, some important points should be addressed and some questions answered prior to publication. The main weakness is the lack of dry biomass data and the use of Petri dishes to determine growth. 

L73: in response “to” salinity: the “to” is missing

Fig. 2: the bars are a little too narrow, it is hard to see the letters. The title for the Y axis should be provided for fig 2b.

L190: Add “it” in front of “has”

L222: reword “superior responses”, be more specific

L256-259: words are missing “germination exposed to”

L262-284: Seeds of different ages (collected in 2016 and 2017) have been used. Could this have affected the results?

L303: the light intensity was very low.

L304:This sentence is confusing: “Spikelets were exposed to treatments for 2 months”.

L322: Why measuring only the length to estimate seedling growth? The authors should explain why they are not presenting the dry biomass. This would have provided very useful information. In a Petri dish growth could have been limited with 25 seeds per Petri plates.  Why not using pots? Or hydroponics systems? Why having so many seeds per dish to measure growth?

Author Response

We thank you for your comments.

Point 1: L73: in response “to” salinity: the “to” is missing

Response 1: “in response salinity” has changed to “in response to salinity”

Point 2: Fig. 2: the bars are a little too narrow, it is hard to see the letters. The title for the Y axis should be provided for fig 2b.

Response 2: We have made the Figure 2 bigger and we provided the tittle for the Y axis for Figure 2b.

Point 3: L190: Add “it” in front of “has”

Response 3: “as has commonly been observed…” has changed to “as it has commonly been observed…”

Point 4: L222: reword “superior responses”, be more specific

Response 4: We have changed the sentence:  "Its higher germination tolerance to salinity, faster germination after being pre-treated at increasing salinities and the presence of seed dormancy help to explain that S. densiflora is able to invade a wide range of habitats along the intertidal gradient [21] including hypersaline saltpans [3]".

Point 5: L256-259: words are missing “germination exposed to”

Response 5: We have changed the sentence:  "As in our study, Iris hexagona Walter and Suaeda aralocaspica (Bunge) Freitag & Schütze growing in high salinities produced seeds that had higher germination rates and speeds of germination when seeds were exposed to different salinity concentrations than seeds produced in low salinity environments"

Point 6: L262-284: Seeds of different ages (collected in 2016 and 2017) have been used. Could this have affected the results?

Response 6: Sorry, we forgot to explain that seeds were sown one month after collection. We have added this information in the sentence: "Before sowing, in September 2017 in case of S. maritima and in December 2016 in case of S. densiflora, the spikelets ..."

Point 7: L303: the light intensity was very low.

Response 7: Yes, is very low compared with the light fluence rate that lands on an open surface in certain month of the year.

Firstly, we use this light fluence rate because it is near that recommended for seed germination studies by the International Seed Testing Association, that is near 20 µmol m-2 s-1 according to C.C. Baskin & J.M. Baskin (2014) Seeds. Ecology, biogeography, and evolution of dormancy and germination 2nd ed. Academic Press, San Diego.

Moreover, from the ecological point of view we must to consider that these plants germinate in fall, and the daily solar radiation at this season in Huelva is around 1-2 kWh m-2. Our seeds received 0.34 kWh m-2, that is around a quarter part, but S. maritima populations are inundated many hours at day, so their seeds are not illuminated at all during the sun hours. And seeds of S. densiflora germinate under shrub canopy shade conditions.

Point 8: L304: This sentence is confusing: “Spikelets were exposed to treatments for 2 months”.

Response 8: We have changed this sentence:

"The dishes were maintained during 2 months under controlled-environmental conditions in a plant grow room, at temperatures between +20 ºC and +25 ºC and a 12h light/12h dark photoperiod. Radiation was provided by fluorescent lamps that produced a photosynthetic photon flux density of 60 μmol m-2 s-1. During this time, germination was recorded every 2 or 3 days. A seed was considered germinated when the coleoptile emerged".

Point 9: L322: 1) Why measuring only the length to estimate seedling growth? The authors should explain why they are not presenting the dry biomass. This would have provided very useful information.

Response 9-1: We consider that measure the growth of cotyledon, first leaf and radicle separately would be interesting because salinity could affect them in a different way, as we have observed in other studies using salts and heavy metals.

For estimating the dry weight of these organs, we need cut them and it probably affects the integrity of the samples, and manipulation of dry organs, especially the radicles, is difficult because they are fragile. So as we suppose dry weight must be related to size, we have used the length of these organs.

2) In a Petri dish growth could have been limited with 25 seeds per Petri plates.  Why not using pots? Or hydroponics systems?

Response 9-2: In dishes there was no interference among seeds or seedlings because they are not large enough to be affected by the neighbour seeds. We always use dishes in these kind of studies because they maintain more homogeneous conditions for all the seeds.

3) Why having so many seeds per dish to measure growth?

Response 9-3: For statistical analysis we need a mean and a standard deviation, and the bigger is the sample the more precise is the result. In case of seedlings we have observed in previous studies that measures often show greater deviations, so we need many seedlings to guarantee significant results from statistical analysis.

Round 2

Reviewer 3 Report

The authors have addressed my comments/questions.